# HumanCrafter: Synergizing Generalizable Human Reconstruction and Semantic 3D Segmentation

**Panwang Pan**[1*‡]**, Tingting Shen**[2*]**, Chenxin Li**[3*]**, Yunlong Lin**[2]**,**
**Kairun Wen**[2]**, Jingjing Zhao**[1]**, Yixuan Yuan**[3‡]

[*] Equal contribution    [‡] Corresponding author
[1]ByteDance, [2]Xiamen University, [3]CUHK
https://paulpanwang.github.io/HumanCrafter

## Abstract

Recent advances in generative models have achieved high-fidelity in 3D human reconstruction, yet their utility for specific tasks (e.g., human 3D segmentation) remains constrained. We propose HUMANCRAFTER, a unified framework that enables the joint modeling of appearance and human-part semantics from a single image in a feed-forward manner. Specifically, we integrate human geometric priors in the reconstruction stage and self-supervised semantic priors in the segmentation stage. To address labeled 3D human datasets scarcity, we further develop an interactive annotation procedure for generating high-quality data-label pairs. Our pixel-aligned aggregation enables cross-task synergy, while the multi-task objective simultaneously optimizes texture modeling fidelity and semantic consistency. Extensive experiments demonstrate that HUMANCRAFTER surpasses existing state-of-the-art methods in both 3D human-part segmentation and 3D human reconstruction **from a single image**.

## 1 Introduction

Reconstructing high-fidelity 3D human representations and understanding human body and clothing attributes are a fundamental challenge in the 3D vision realm. The philosophy can unlock many novel and practical downstream applications, such as semantic-guided 3D reasoning, context-aware human behavior analysis, and interactive semantic editing. This capability further facilitates immersive augmented and virtual reality (AR/VR), character stylization, and cinematic production.

In light of advances in Neural Radiance Fields [1], previous attempts [2, 3, 4] have synthesized high-quality human novel views. Nevertheless, implicit representations are typically computationally intensive, as they rely on dense point querying in 3D space. Recent advances in 3D Gaussian Splatting (3DGS) [5] have provided real-time rendering for reconstructing high-quality explicit human models, which however rely on dense multi-view images [6, 7] or monocular video input [8, 9, 10] and time-consuming per-subject optimization processes, limiting their stability and feasibility in downstream applications. With the emergence of large reconstruction models [11], recent advances can directly generalize the regression of 3D representations [12, 13, 14, 15] thanks to the prevalence of large-scale 3D object datasets [16]. However, in the specific task of human reconstruction, these works collapse and fail to produce faithful and consistent novel views, primarily due to the scarcity of 3D human datasets and a lack of human prior knowledge as inductive bias in model design. Recent pioneering studies on human reconstruction [17, 18] enable generalizable and robust synthesis under sparse-view settings. However, the crucial requirement for semantic 3D segmentation is currently hindered by the lack of expansive, well-labeled 3D human downstream task datasets. One workaround is to first reconstruct and then leverage recent advances in 2D human visual foundation models [19]. This paradigm can result in extensive processing times and substantial engineering efforts. Furthermore,

39th Conference on Neural Information Processing Systems (NeurIPS 2025).

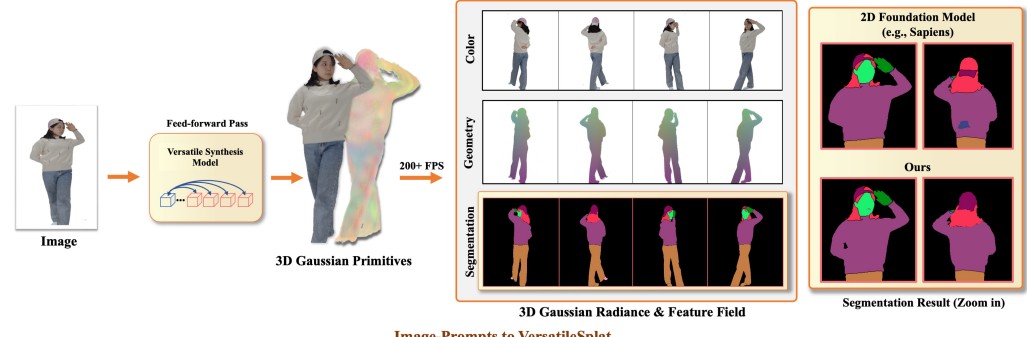

Figure 1: We introduce HUMANCRAFTER, a unified framework for simultaneous human 3D reconstruction and body-part segmentation from single images. HUMANCRAFTER introduces explicit 3D Gaussian VersatileSplats, showcasing enhanced performance over foundation models in delivering 3D-consistent segmentation outcomes. This breakthrough offers significant advantages for downstream applications.

2D operations cannot maintain 3D consistency and coherence across different viewpoints. The two-stage pipeline faces challenges including inconsistencies, high computational costs, and scalability issues, which hinder its robustness and efficiency. We hold the belief that *the best way to understand something is to reconstruct it*.

We introduce a unified synthesis model that revolutionizes 3D reconstruction and editing through innovative view synthesis and semantic understanding. Our model takes a single image as input and generates explicit 3D Gaussian Splats enriched with semantic features, enabling human 3D reconstruction and body-part segmentation, while seamlessly enabling the 3D editing task and integrating into VR devices, as illustrated in Figure 1. Our approach builds upon the incorporation of tailored human priors and the aggregation of multi-view features with camera embeddings using Transformers. Specifically, we translate the set of aggregated features to pixel-aligned 3D Gaussians as initialized geometry. We unleash a pre-trained 2D model [20] into a 3D consistent feature field and establish a weighting mechanism to propagate into multi-view, addressing the scarcity of labeled 3D semantic data. Finally, by jointly training on the constructed 3D segmentation datasets, which consist of 40,000 images from 2,500 human scans, HUMANCRAFTER enables novel view rendering and segmentation to mutually benefit from each other's task. In summary, our contributions can be summarized as follows:

- We are first to introduce a unified 3D human representation and a holistic framework that addresses versatile novel-view synthesis in a feed-forward pass, allowing two independent tasks to mutually benefit.

- HUMANCRAFTER leverages geometric human priors and the attention mechanism to effectively bypass labor-intensive and computationally expensive steps, establishing a new paradigm in the realm of human foundation models.

- Experiments demonstrate that HUMANCRAFTER exhibits superior photorealistic 3D human Reconstruction and human-part segmentation capabilities, surpassing many state-of-the-art baselines simultaneously with real-time rendering.

- Experiments on in-the-wild images demonstrating HUMANCRAFTER's strong generalizability to diverse image, and facilitating the potential real-world applications.

## 2 Related Work

### 2.1 Synergizing Reasoning with 3D Reconstruction.

2D human-centric models [21, 22, 19] exhibit remarkable performance through the advent of 2D foundation models and vision transformers [23, 20] and curated datasets (e.g., Humans-300M [19]). Yet, these cutting-edge methods cannot achieve simultaneous 3D modeling and coherent segmentation due to lack 3D constraints. A crucial requirement for 3D reasoning is currently hindered by the paucity of extensive, well-annotated multi-view image datasets. Conversely, early studies like Semantic NeRF [24] and Panoptic Lifting [25] successfully embedded segmentation network semantic data into 3D scenes. 3DGS with Feature Field [26, 27, 28, 29] emerged as a prominent joint training approach for 3DGS and multiple prediction tasks. Recent works [30, 31] enable training-free 2D feature lifting to 3D paradigm for large scenes. However, the human-specific domain remains largely unexplored, despite its potential applications in areas such as human editing, gaming, and film production.

### 2.2 Human Gaussian Splatting.

Recent advances [6, 32, 7] optimize photo-realistic animatable 3DGS from temporal multi-view images. In particular, HiFi4G [33] combines 3DGS with a dual-graph mechanism to maintain spatial-temporal coherence, ASH [34] employs mesh UV parameterization for real-time rendering, and Animatable Gaussians [35] utilizes StyleUNet [36] and 3DGS for high-fidelity animatable avatars. For monocular video input [8, 37, 9, 38, 39, 40, 41, 42, 43, 44], human templates and Linear Blend Skinning are commonly adopted, necessitating per-instance optimization of 3DGS in a canonical space. For sparse-view scenarios, GPS-Gaussian [17] estimates depth maps from two-view stereo and unprojects them to pixel-wise 3D Gaussians. [45] further introduces a regularization term and an epipolar attention mechanism to preserve geometry consistency between source views. EVA-Gaussian [46] employs a recurrent feature refiner to address artifacts. [18] utilizes human templates for multi-scaffold photorealistic and accurate view rendering. For single image input, [47] incorporates generative diffusion models to predict triplane NeRF and followed by a feed-forward reconstruction model. Recent advances [48, 49, 50, 51] predicts the 3D outputs from a single input image in a generalizable manner through the combination of 2D Diffusion and well-conceived human priors. Notably, Human-3Diffusion [51] introduces a groundbreaking single-image-to-3DGS framework that synergizes diffusion models with 3D reconstruction to create highly consistent 3D avatars. Our work further enhances the synergy between 3D reconstruction and semantic segmentation. Our unified pipeline utilizes efficient multi-view geometric aggregation and a refined Transformer module to significantly improve accuracy, real-time rendering, and multi-task consistency.

### 2.3 Generalizable Reconstruction Transformer.

GINA-3D, LRM and TripoSR [52, 53, 54] demonstrate that feed-forward Transformers trained on large-scale datasets [16, 55] are capable of generating 3D models in a generalizable manner by reconstructing triplane features for volumetric rendering. Real3D [56] further proposes a self-training reconstruction framework that capitalizes on both synthetic and large-scale real-world datasets. Instant3D [57] proposes a two-stage pipeline, which first generates a sparse set of four posed images, and then directly regresses the Neural Radiance Fields (NeRF) [58]. [59] integrates efficient 3DGS with triplane representations, where point clouds inferred from input images query triplane features to decode 3DGS attributes. CRM, MeshLRM, and InstantMesh [60, 61, 62] replace the triplane NeRF with FlexiCubes [63] as outputs, incorporating differentiable mesh extraction and rendering for potential applications. LGM, GRM, and GS-LRM [64, 65, 66] predict pixel-wise 3DGS parameters [67, 68] from multiple images, enabling scalable reconstruction frameworks. However, such methods lack human-specific priors as tailored inductive biases in Transformers, failing to synthesize faithful novel views, especially reconstructing human body details.

## 3 Method

**Preliminary of 3D Gaussians Splatting.** 3DGS [5] formulates the 3D representation as a collection of $N$ Gaussian primitives $\{\boldsymbol{G}_p\}_{p=1}^{N}$. Each $\boldsymbol{G}_p$ is characterized by an opacity $\boldsymbol{\sigma}_p \in \mathbb{R}$, a mean location $\boldsymbol{\mu}_p \in \mathbb{R}^3$, a scaling factor $\boldsymbol{s}_p \in \mathbb{R}^3$, an orientation quaternion $\boldsymbol{q}_p \in \mathbb{R}^4$, and a color feature $\boldsymbol{c}_p \in \mathbb{R}^C$

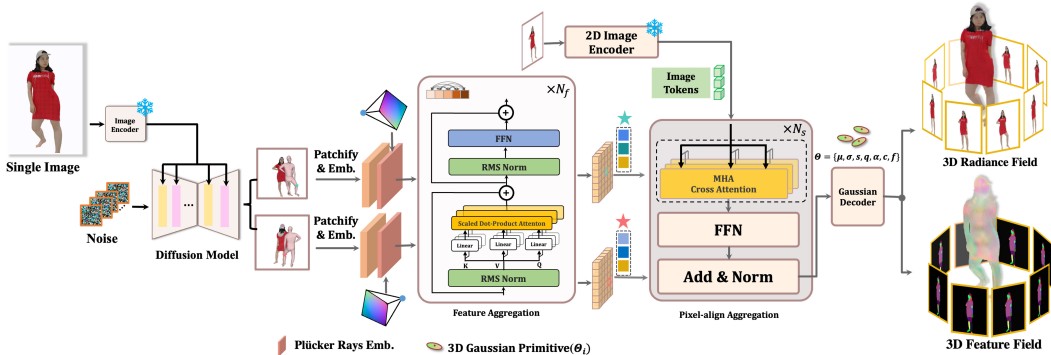

Figure 2: **The network architecture of HUMANCRAFTER**. The proposed method fully utilizes 2D diffusion priors and human body geometry features to regress pixel-aligned point maps via a generic Transformer (Sec. 3.1). Subsequently, another Transformer (Sec. 3.2) employs an attention mechanism to produce a set of semantic 3D Gaussians that encapsulate geometric, appearance, and semantic information. The entire pipeline is trained in an end-to-end manner by minimizing a loss function (Sec. 3.3) that compares the predicted outputs against ground truth data and rasterized label maps from novel viewpoints.

are maintained for rendering, where spherical harmonics (SH) can model view-dependent effects. Specifically, each Gaussian can be formulated as: $G_p(\mathbf{x}) = \exp\left(-\frac{1}{2}(\mathbf{x} - \boldsymbol{\mu}_p)^\top \boldsymbol{\Sigma}_p^{-1}(\mathbf{x} - \boldsymbol{\mu}_p)\right)$.

The rendering process involves projecting the 3D Gaussians onto the image plane as 2D Gaussians and performing alpha blending for each pixel in front-to-back depth order, thereby determining the final color, opacity, and depth maps.

**Pipeline Overview.** Given a single RGB image $\mathbf{I} \in \mathbb{R}^{\mathbf{H} \times \mathbf{W} \times 3}$, our objective is to jointly synthesize multiple properties at novel views using the multi-task labels from the given source views without any optimization or fine-tuning. To accommodate versatile labels, we extend the 3D Gaussians $G_p$ to construct a feature field, enabling the rendering of feature maps through a differentiable rasterizer.

To this end, we firstly propose a purely Transformer (Sec. 3.1) to aggregate multi-view input images, followed by a attention mechanism (Sec. 3.2) to facilitate downstream tasks. Furthermore, we propose a multi-task loss (Sec. 3.3) to achieve high-fidelity texture rendering. An overview of the model architecture is illustrated in Figure 2.

## 3.1 Human Prior for Feature Aggregation

**Image Diffusion Prior.** We leverage the pre-trained 2D diffusion model SV3D [69] as appearance prior. For the input image $\mathbf{I}_0$, we leverage a CLIP image encoder [70] to obtain the image embedding $\mathbf{c}$ as conditions. Subsequently, we progressively denoises gaussian noises into temporal-continuous $\mathbf{N}$ multi-view images by a spatial-temporal UNet $D_\theta$ [71]. In this work, we also employ the SMPL [72] as the 3D human parametric model to guide the Feature Aggregation. Inspired by [73, 74], we render the side-view normal images as a guide, which are then concatenated with the input image and corresponding Plücker embeddings [75] along the channel dimension, resulting in dense pose-conditioned images. These pose-conditioned images are divided into non-overlapping patches [23] and mapped to $d_1$ dimensional patch tokens $\mathbf{F}_i \in \mathbb{R}^{(h \times w) \times d_1}$ by a Linear layer, where $h = \mathbf{H}/\mathbf{P}$ and $w = \mathbf{W}/\mathbf{P}$ represent the height and width of the feature map, respectively, and $\mathbf{P}$ denotes the patch size.

**Cross-view Attention Module.** For feature interaction within patch tokens, we utilize $\mathbf{N}_f$ layers of Grouped Query Attention blocks [76] with RMS Pre-Normalization, GELU activation, and feed-forward network. These features are subsequently mapped into the location of 3D Gaussian Splatting. To accurately model the positioning of Gaussians, we predict the depth map $\mathbf{D}_i \in \mathbb{R}^{\mathbf{H} \times \mathbf{W}}$ for each input image and additional 3D positional offset $\boldsymbol{\Delta}_i \in \mathbb{R}^{\mathbf{H} \times \mathbf{W} \times 3}$. A pixel located at $(u, v)$ in the image $\mathbf{I}_i$ is unprojected from the image plane onto the 3D position $\boldsymbol{\mu}_p \in \mathbb{R}^3$ by unproject function

$\mathbf{\Pi}^{-1}(\mathbf{K}; \mathbf{R}_i; \mathbf{t}_i)$:

$$\mathbf{\Pi}^{-1}[u, v] := \mathbf{R}_i^\top \mathbf{K}^{-1}\mathbf{D}_i[u, v] - \mathbf{t}_i + \mathbf{\Delta}[u, v] \tag{1}$$

where, $\mathbf{D}_i[u, v]$ represents the depth value at pixel $(u, v)$ in the $i$-th view's depth map; $\mathbf{R}_i$ is the rotation matrix of the $i$-th view, $\mathbf{K}$ is the camera intrinsic matrix, and $\mathbf{t}_i$ is the translation vector of the $i$-th view. The proposed Transformer architecture leverages feature aggregation to synergistically exploit human geometry priors and the information embedded within input images, effectively bridging the 2D feature space and 3D coordinates to deliver enriched geometric representations.

## 3.2  Self-Supervised Model as Inductive Bias

We employ the DINOv2 `vits14-with-registers` [20, 77] as 2D Image Encoder, which has been proven effective for 3D scene segmentation [78] and diverse downstream tasks [79]. We `freeze` the network parameters and extract features from the input image, denoted as $\mathbf{f}_i \in \mathbb{R}^{(h \times w) \times d_2}$, where $d_2$ is the dimension of the image encoder. Furthermore, as the feature aggregation stage for the posed images has already learned the relevance between each position token, we directly extract the attention weights from the previous Transformer and perform a weighted combination of the features which is independent of the feature dimension. We leverage the tailored inductive biases learned by the preceding Transformer across the hierarchical Transformer blocks. Specifically, for each Transformer block, the feature-wise similarity is extracted by the dot product of $\mathbf{Q}(\mathbf{F}_i)$ and $\mathbf{K}(\mathbf{F}_i)$ corresponding to each $\mathbf{V}(\mathbf{f}_i)$.

$$CrossAttn(\mathbf{f}_i) := SoftMax\left(\frac{\mathbf{Q}(\mathbf{F}_i)\mathbf{K}(\mathbf{F}_i)^\top}{\sqrt{d_k}} + \mathbf{B}\right)\mathbf{f}_i \tag{2}$$

where, $d_k$ presents the dimension of $\mathbf{F}_i$, $\mathbf{B}$ is the relative position bias. Ultimately, from each output token $\tilde{\mathbf{f}}_i \in \mathbb{R}^{(h \times w) \times d_2}$, we decode the attributes of pixel-aligned Gaussians $\boldsymbol{G}$ in the corresponding patch using a convolutional layer with a $1 \times 1$ kernel. The final pixel color $\mathbf{c}$ is calculated by blending $\mathbf{N}$ ordered Gaussians overlapping the pixels via the following rendering function: $\mathbf{c} = \sum_{i=1}^{\mathbf{N}} c_i \boldsymbol{\sigma}_i \prod_{j=1}^{i-1}(1 - \boldsymbol{\sigma}_j)$. This equation efficiently models the contributions of each Gaussian to the pixel's final appearance, accounting for their transparency and layering order. To facilitate body-part semantic 3D representation, inspired by [26], we augment the 3D Gaussians Splats with a learnable semantic feature embedding (denoted as 3D Gaussians primitives) and rasterize onto the 2D image plane by blending Gaussians that overlap with each pixel using a feature rendering function. This implies that the novel view synthesis task and human-part segmentation task share the same 3D Gaussian parameters, where $\mathbf{N}$ is the number of Gaussian primitives participating in the blending: $f = \sum_{i=1}^{\mathbf{N}} \mathbf{M}\tilde{\mathbf{f}}_i \boldsymbol{\sigma}_i \prod_{j=1}^{i-1}(1 - \boldsymbol{\sigma}_j)$. where $f$ indicates the final rasterized feature embedding on the image plane, and $\tilde{\mathbf{f}}_i$ represents the semantic

## 3.3  Multi-task Training Objective

The 3D reconstruction loss function is designed to minimize the rendering loss $\mathcal{L}_{render}$ for novel viewpoints. The loss for $k_v$ rendered multi-view images is defined as:

$$\mathcal{L}_{render} := \mathbb{E}[\mathcal{L}_{mse}(\hat{\mathbf{I}}_i, \mathbf{I}_i) + \lambda_m \mathcal{L}_{mask}(\hat{\mathbf{M}}_i, \mathbf{M}_i) + \lambda_p \mathcal{L}_{LPIPS}(\hat{\mathbf{I}}_i, \mathbf{I}_i)] \tag{3}$$

where $\mathbf{I}_i$ and $\hat{\mathbf{I}}_i$ denote the ground-truth images and rendered images via 3D Gaussian Splatting, respectively. $\mathbf{M}_i$ and $\hat{\mathbf{M}}_i$ represent the original and rendered foreground masks. $\mathcal{L}_{mse}$ measures the mean squared error loss, and $\mathcal{L}_p$ measures the perceptual loss [80]. $\lambda_m$ and $\lambda_p$ are hyperparameters employed for balancing the respective loss terms. Building upon prior work [31], the loss function for the feature field is minimized during training by utilizing rasterized feature maps on novel views $\mathbf{f}$ and directly inferred feature maps using ground truth images on new views $\hat{\mathbf{f}}$, thereby facilitating the learning of blending weights for consistent semantic field regression. $\mathcal{L}_{dist} = 1 - \texttt{CosSim}(\hat{\mathbf{f}}, \mathbf{f}) = 1 - \frac{\hat{\mathbf{f}} \cdot \mathbf{f}}{\|\hat{\mathbf{f}}\|\|\mathbf{f}\|}$, where $\texttt{CosSim}(\cdot, \cdot)$ denotes the cosine similarity between the predicted and ground truth feature maps, which serves as a distance metric to be minimized during training. Finally, we construct a semantic segmentation dataset for fine-tuning. We employ 28 classes for body-part segmentation,

along with the background class, following [19]. We jointly train the network on all tasks using differentiable rendering. Our model can be optimized in an end-to-end manner:

$$\mathcal{L}(\Theta) := \mathop{\mathbb{E}}_{i \in \{1,...k_v\}} [\mathcal{L}_{\text{render}} + \lambda_{\text{dist}} \cdot \mathcal{L}_{\text{dist}}(\mathbf{f}_i, \hat{\mathbf{f}}_i)] + \lambda_{\text{seg}} \cdot \mathop{\mathbb{E}}_{j \in \{1,...k_s\}} [\mathcal{L}_{\text{CE}}(\mathbf{S}_j, \hat{\mathbf{S}}_j)] \quad (4)$$

where $\Theta$ is the model parameters, $\mathbf{S}_i$ and $\hat{\mathbf{S}}_j$ denote the annotated and predicted semantic segmentation, respectively. $L_{\text{CE}}$ represents the `Cross Entropy Loss`, and we enforce the predicted score to align with the ground truth viewpoint. Since we only have a small number of annotated viewpoints, $L_{\text{CE}}$ is added only when $j \in \{1,...k_s\}$. The Aggregation mechanism is independent of features. Therefore, we can leverage a 2D pretrained model to supervise the generation of rasterized novel views using $\mathcal{L}_{\text{dist}}$, and allow the two tasks to benefit each other through $\mathcal{L}_{\text{CE}}$.

## 4 Experiments

We conducted experiments on the following datasets to evaluate the results of 3D human reconstruction and segmentation. The 2D diffusion model takes approximately 6 seconds (with the number of views set to 2) to generate multi-view latent features, while the subsequent reconstruction stage takes only about 0.2 seconds.

**Datasets.** (1) THuman2.1 Dataset [81] contains approximately 2500 human scans. Specifically, we select 2300 scans for training and the rest for evaluation. (2) 2K2K Dataset [82] includes 2000 human scans. Similarly, we select 1500 scans for training and the rest for evaluation. (3) Human MVImageNet [83] approximately comprises 4000 identities and 8000 outfits, which provide the rich multi-perspectives. Consistent with the protocol established in [48], we utilize PIXIE [84] as the SMPL parameter estimator, strategically placing 36 cameras across three hierarchical levels to capture full-body, upper-body, and facial views, with all renderings resolution of $512 \times 512$ pixels.

**Curated Dataset.** We randomly select 500 scans from the training dataset and annotate 8 semantic segmentation maps for each scan. More details are provided in Appendix A.1.

**In-the-wild Dataset.** To assess the model's generalizability under challenging conditions, we construct a test dataset from Internet-sourced images. These images encompass diverse human poses, identities, and camera viewpoints.

**Training Details.** The hyperparameters $\lambda_{\text{mask}}$, $\lambda_{\text{p}}$ are set to 1 and 0.1 in this paper. The hyperparameters $\lambda_{\text{dist}}$ and $\lambda_{\text{dist2}}$ are both set to 0.5. We use the AdamW optimizer with $\beta_1 = 0.9$ and $\beta_2 = 0.95$, and a weight decay of 0.05 is applied to all parameters except those in the LayerNorm layers. A cosine learning rate decay scheduler is employed, with a linear warm-up of 2,000 steps. The peak learning rate is set to $4 \times 10^{-4}$. The training process is divided into two stages: the model is trained for 80K iterations at $256 \times 256$ resolution and then fine-tuned for an additional 20K iterations at $512 \times 512$ resolution. Please refer to Appendix A.2 for more detailed procedural insights.

### 4.1 Evaluation of Human 3D Segmentation

**Comparisions.** The semantic segmentation is evaluated by class-wise intersection over union (mIoU) and average pixel accuracy (mAcc) on novel views as metrics. To provide a more comprehensive evaluation of the algorithm's 3D consistency, we compare HUMANCRAFTER against the state-of-the-art LSM [31] baseline. Additionally, we benchmark our approach against the 2D state-of-the-art

Table 1: Comparison with feed-forward Human 3D Segmentation methods on 2K2K dataset.

| Method | 2K2K [82] | | | | | Runtime |
|---|---|---|---|---|---|---|
| | mIOU ↑ | Acc. ↑ | PSNR↑ | SSIM ↑ | LPIPS ↓ | |
| *Two GT Input Views* | | | | | | |
| LSM* [31] | 0.724 | 0.873 | 23.811 | 0.892 | 0.053 | **108 ms / object** |
| Sapiens [19] | 0.823 | 0.904 | N/A | N/A | N/A | 640 ms / frame |
| Ours | 0.840 | 0.925 | 24.786 | 0.937 | 0.022 | 126 ms / object |
| *Single View* | | | | | | |
| Human3Diffusion [51] + Sapiens [19] | 0.781 | 0.851 | 21.832 | 0.891 | 0.069 | 23.21 s / object |
| Ours | **0.801** | **0.882** | **23.489** | **0.916** | **0.045** | **6.24 s / object** |

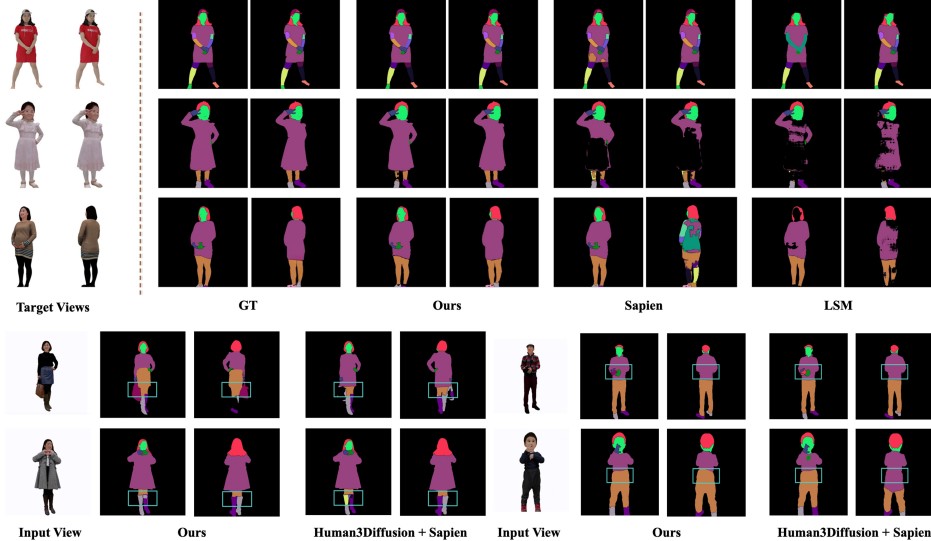

Figure 3: Qualitative Results and Comparisons on Human 3D Segmentation on THuman2.1 and 2K2K datasets. HUMANCRAFTER achieves the best precise segmentation results in terms of 3D consistency.

human segmentation algorithm proposed in [19], which is trained on large-scale human datasets. LSM* is trained using 3D human scans and 2D semantic segmentation maps, and it takes two images as input to ensure a fair comparison. As illustrated in Table 1, HUMANCRAFTER outperforms the state-of-the-art baselines in terms of segmentation accuracy, while exhibiting comparable reconstruction times. To enhance single-image reconstruction, we employ a two-stage approach using cutting-edge algorithms [51] as a baseline. As shown in Table 1, HUMANCRAFTER surpasses baselines in segmentation accuracy while maintaining comparable reconstruction times. Furthermore, Figure 3 demonstrates the superior segmentation quality achieved by proposed method.

## 4.2 Evaluation of 3D Human Reconstruction

**Comparisions.** The quantitative assessment leverages metrics such as PSNR, SSIM [87], and VGG-LPIPS [80] to comprehensively analyze the rendering fidelity. Inspired by Diffsplat [], multi-view consistency is evaluated through COLMAP reconstructed point number [88]. We compare our approach with state-of-the-art methods. These include advanced reconstruction-based techniques for single-image-conditioned generation: LGM [64], GRM [85], InstantMesh [62], Lara [86], Human3Diffusion [51], and PSHuman [49]. For the single-view setting, our method can either replicate the same input or utilize existing multi-view diffusion models available on the shelf to introduce 2D generative priors and achieve better results, as demonstrated in the last row of Table 2. The single image-conditioned generation performance on the THuman 2.1 and 2K2K datasets is evaluated in Table 2, with Qualitative results on challenging scenarios (e.g., far camera viewpoints, complex human poses, and loose clothing) are presented in Figure 4 and Figures 7 in the Appendix. HU-MANCRAFTER demonstrates superior performance compared to state-of-the-art baselines in both

Table 2: Comparison with feed-forward 3D reconstruction methods at a resolution of $512 \times 512$.

| Method | THuman2.1 [81] | | | | 2K2K [82] | | | |
|---|---|---|---|---|---|---|---|---|
| | PSNR↑ | SSIM↑ | LPIPS↓ | #Points ↑ | PSNR↑ | SSIM↑ | LPIPS↓ | #Points ↑ |
| LGM† [64] | 20.106 | 0.859 | 0.196 | 502.05 | 21.685 | 0.850 | 0.166 | 694.84 |
| GRM† [85] | 20.503 | 0.868 | 0.141 | 602.46 | 21.496 | 0.858 | 0.171 | 722.95 |
| InstantMesh [11] | 19.997 | 0.875 | 0.128 | 1803.28 | 21.983 | 0.865 | 0.118 | 2028.30 |
| LaRa [86] | 18.120 | 0.840 | 0.207 | 3035.43 | 19.113 | 0.860 | 0.207 | 4139.94 |
| SiFU [73] | 20.164 | 0.842 | 0.088 | 4500.68 | 21.698 | 0.904 | 0.084 | 4560.18 |
| PSHuman [49] | 20.853 | 0.862 | 0.076 | 5321.23 | 21.932 | 0.892 | 0.076 | 4734.23 |
| Human3Diffusion [51] | 22.164 | 0.872 | 0.063 | 5123.52 | 22.323 | 0.882 | 0.053 | 5134.23 |
| HumanCrafter | **23.186** | **0.907** | **0.046** | **5744.96** | **23.489** | **0.916** | **0.045** | **6453.23** |

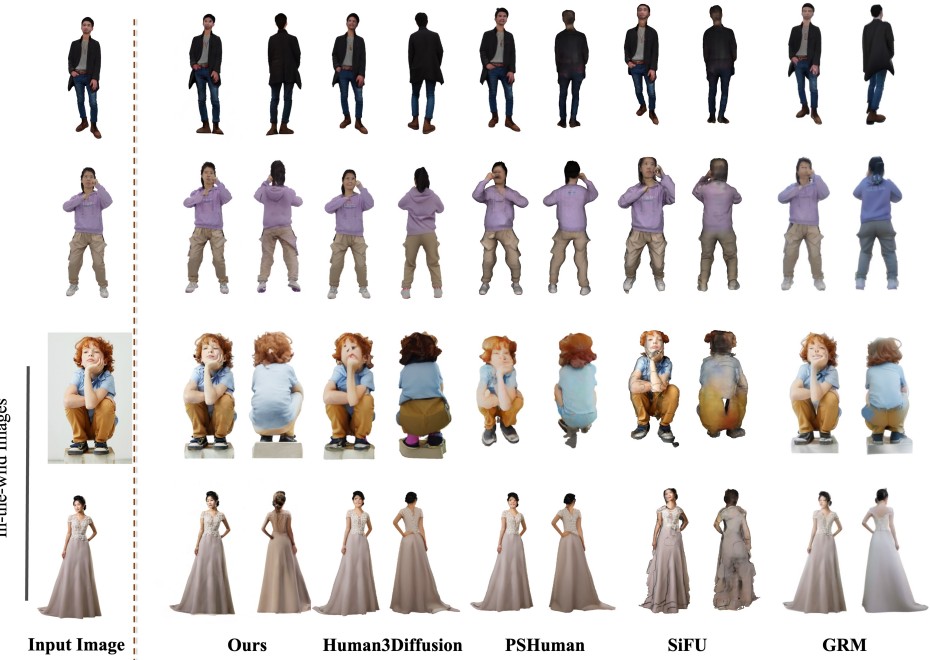

In-the-wild Images

| Input Image | Ours | Human3Diffusion | PSHuman | SiFU | GRM |

Figure 4: Novel-view images rendered by HUMANCRAFTER and the state-of-the-art baselines on various datasets. Our method achieves the highest rendering quality. Please refer to the zoomed-in regions for details.

rendering quality and 3D consistency metrics. Notably, LGM† and GRM† have been fine-tuned on our dataset to ensure a fair comparison.

## 4.3 Ablation Studies

we carefully investigate the effectiveness of human prior and each design choice in this subsection.

**The Effect of Human Parametric Model.** As shown in Table 3, the human geometric prior benefits significantly from the reconstruction model, particularly when $k_v = 1$. Additionally, in the context of human body reconstruction tasks, rendering SMPL normals can provide richer geometric cues compared to relying on coordinate or rendered depth maps. As the number of input viewpoints $k_v$ increases, the reconstruction model effectively resolves geometric ambiguities through stereo matching, reducing its dependence on Human Pose Estimation. The statistics in Table 3 validate HUMANCRAFTER consistently outperforms other variants in terms of image quality and fidelity.

**Model Design Choices.** As demonstrated in Table 4 (b)-(d), the experiments confirm the efficacy of each crucial design decision. **i)** The HUMANCRAFTER model, in the absence of Plucker embeddings and relies solely on the SMPL-based geometric prior, demonstrates a marginal decline in performance. **ii)** When integrating the pre-trained MAE model, denoted as `ViT-s` [89], in place of the DINOv2 model, HUMANCRAFTER exhibits a slight performance decrease. This alteration is attributed to

Table 3: Ablation study of human geometry prior on 2K2K dataset.

|  | $k_v$ | PSNR ↑ | SSIM ↑ | LPIPS ↓ | #Param. ↓ |
|---|---|---|---|---|---|
| w/o SMPL | 1 | 22.203 | 0.890 | 0.064 | 41.2M |
| + Depth | 1 | 23.103 | 0.901 | 0.0270 | 42.4M |
| + Coord. | 1 | 23.324 | 0.917 | 0.047 | 42.6M |
| + Normal (Ours) | 1 | 23.489 | 0.916 | 0.045 | 42.4M |
| w/o SMPL | 2 | 23.570 | 0.913 | 0.034 | 41.2M |
| with SMPL (Ours) | 2 | 24.786 | 0.937 | 0.022 | 42.6M |

Table 4: Ablation of model and objective design on 2K2K dataset.

|  | PSNR ↑ | SSIM ↑ | LPIPS ↓ |
|---|---|---|---|
| (a) Full Model (Ours) | **23.489** | **0.916** | **0.045** |
| (b) w/o Cam. Emb. | 23.327 | 0.903 | 0.048 |
| (c) w/o DINOv2 | 22.025 | 0.891 | 0.055 |
| (d) w/o Pixel-align Aggregation | 21.183 | 0.891 | 0.067 |
| (e) w/o $\mathcal{L}_{\text{dist}}$ | 22.464 | 0.896 | 0.055 |
| (f) w/o $\mathcal{L}_{\text{CE}}$ | 23.223 | 0.901 | 0.051 |

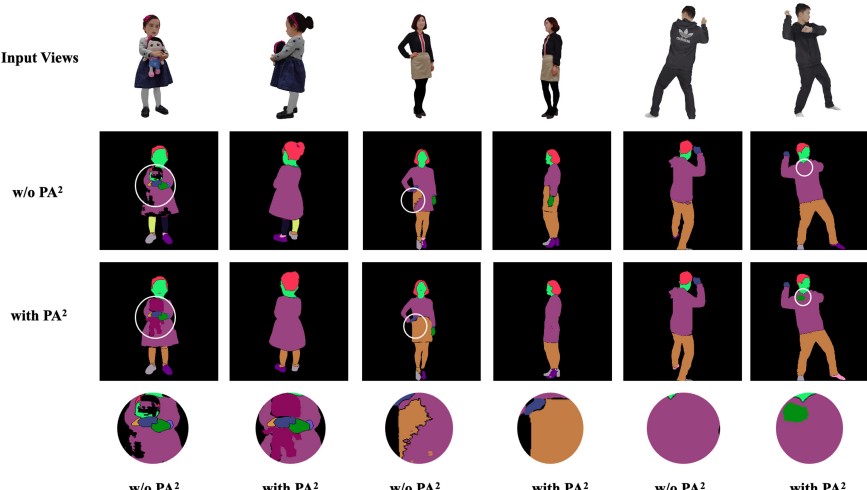

Figure 5: **Ablation of Pixel-Align Aggregation**. HUMANCRAFTER with PA$^2$ can leverage knowledge learned from novel-view synthesis task and incorporate a pre-trained 2D model, thereby boosting semantic tasks.

the superior ability of the selected pre-trained model to establish correspondences among the input view images, a crucial factor for pixel-aligned aggregation. **iii)** Furthermore, HUMANCRAFTER is solely enhanced with the Feature Aggregation Module akin to the current LGM methodology, and the learned features are directly forwarded through the GS-Decoder, a notable decline in performance is observed, as illustrated in Table 4 (d) and Figure 5 in the Appendix. This result underscores the efficacy of the Pixel-align Aggregation Module.

**The Effect of Loss Functions.** As demonstrated in Table 4 (e)-(f), without the LPIPS loss, novel view renderings are susceptible to blurriness and unnatural generations, leading to a slight performance decline. The integration of the distillation loss enhances 3D view consistency. Similarly, akin to LSM, the addition of the semantic distillation loss illustrates that integrating the human segmentation task enhances the performance of novel view synthesis.

## 4.4 Applications: Human Editing and Immersive Exploring

Figure 1 and Figure 6 illustrate the potential scenarios enabled by the proposed model: **(1) Text to VersatileSplats:** ControlNet [90] is used to produce a human image with the human mask and text prompts. Subsequently, HUMANCRAFTER is employed to reconstruct a 3D human from the generated image. **(2) 3D Consistency Editing:** HUMANCRAFTER's 3D coherence and precise semantic masks are employed to direct the 3D human editing process, supported by a FLUX-based inpainting model [91]. **Immersive Exploring:** HUMANCRAFTERshowcases high efficiency, enabling real-time end-to-end 3D modeling. Following the generation of edited 3DGS primitives for the provided input views, seamless integration into VR devices.

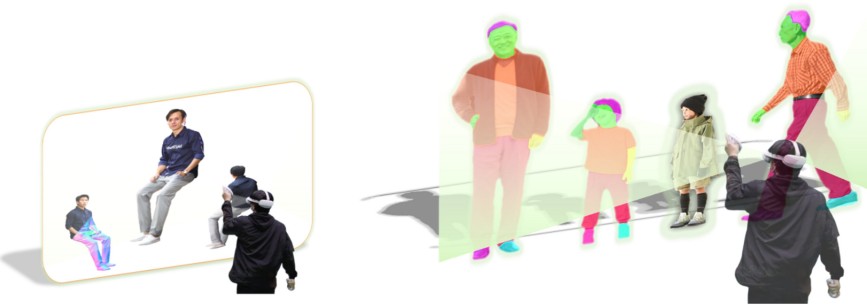

Figure 6: We demonstrate the generalizability of HUMANCRAFTER with in-the-wild images in challenging scenarios.

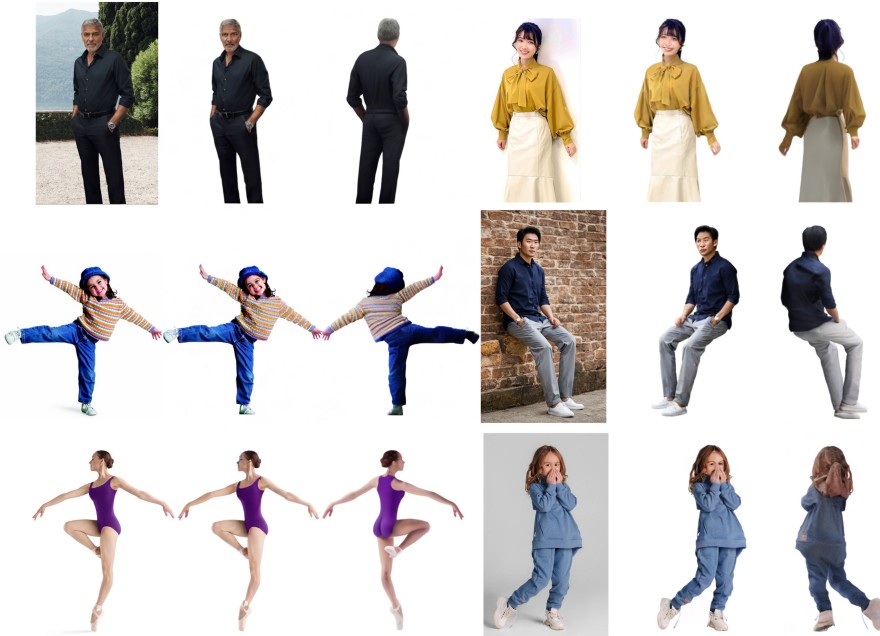

Figure 7: We demonstrate the generalizability of HUMANCRAFTER with in-the-wild images in challenging scenarios.

As depicted in Figure 4.3, we validate that Pixel-Align Aggregation effectively utilizes information from the task of synthesizing new viewpoints. As illustrated in Figure 6, a potential application of our proposed model is its combination with the existing Text-to-Image (T2I) inpainting Diffusion model, such as FLUX.1 [91]. Notably, the 3D Gaussian primitives we generate can seamlessly integrate into VR devices. We demonstrate this through the application of watching a high-fidelity virtual concert using the PICO 4 Ultra VR headset. As depicted in Figure 7, we showcase the generalizability of our model using in-the-wild images in challenging scenarios.

## 5 Conclusion

We have introduced HUMANCRAFTER, a unified framework for 3D human reconstruction and understanding. First, we adopt tailored human priors and aggregate multi-view images from a 2D diffusion model and camera embedding features in a Transformer. Second, we translate the set of aggregated features to pixel-aligned 3D Gaussians as initialized geometry. We extend a 2D pre-trained model into a 3D consistent feature field and establish a weighting mechanism to propagate into multi-view. Extensive experiments demonstrate that HUMANCRAFTER surpasses existing methods in terms of novel view synthesis quality and downstream task performance while exhibiting robustness in complex scenarios.

**Broader Impacts**    HUMANCRAFTER allows users to generate 3D human models tailored to their specific inputs, enabling a broad spectrum of downstream applications, such as AR/VR Chat, 3D cinematography, and 3D editing. However, this capability also presents potential ethical challenges, including privacy violations and racial biases. To mitigate these risks, it is imperative to establish robust ethical guidelines and enforce legal regulations.

**Limitations and Future Works.**    Building on the significant acceleration our method provides for semantic 3D human reconstruction, a compelling avenue for future work is its extension to dynamic 4D scene generation with Gaussian representations [92]. Furthermore, by leveraging web-scale human datasets and relying solely on 2D supervision, extensive real-world video datasets could further unlock the potential of HUMANCRAFTER. Moreover, there is potential for misuse, such as the arbitrary distribution of digital assets. These risks can be mitigated by embedding watermarks into the 3D assets [93, 94].

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

# A    More Implementation Settings

We provide comprehensive implementation details in this section to facilitate the reproducibility of our work. Specifically, in Section. A.1 , we provide the details of how to construct semantic segmentation data. Section. A.2, we provide details about training details. In Section. B, we offer further explanation of the implementation for HUMANCRAFTER.

## A.1    Constructed Dataset Details

**Interactive Annotation.** Human-part Semantic Segmentation aims to classify pixels in the input image $\mathbf{I}_i$ into $\mathbf{N}_{class}$ categories while ensuring 3D consistency. Following Sapiens [19], we construct a dataset with $\mathbf{N}_{class} = 28$ (27 body parts and one background class). The class names are as follows: 'Background', 'Apparel', 'Face_Neck', 'Hair', 'Left_Foot', 'Left_Hand', 'Left_Lower_Arm', 'Left_Lower_Leg', 'Left_Shoe', 'Left_Sock', 'Left_Upper_Arm', 'Left_Upper_Leg', 'Lower_Clothing', 'Right_Foot', 'Right_Hand', 'Right_Lower_Arm', 'Right_Lower_Leg', 'Right_Shoe', 'Right_Sock', 'Right_Upper_Arm', 'Right_Upper_Leg', 'Torso', 'Upper_Clothing', 'Lower_Lip', 'Upper_Lip', 'Lower_Teeth', 'Upper_Teeth', and 'Tongue'.

To accelerate the manual annotation process, we utilize the Segment Anything Model (SAM) [95] for assisted labeling. The dataset construction process is illustrated in Figure 8. In (a), we show the instance data and segmentation pipeline, and in (b), we demonstrate how to accelerate annotation using Segment Anything Model.

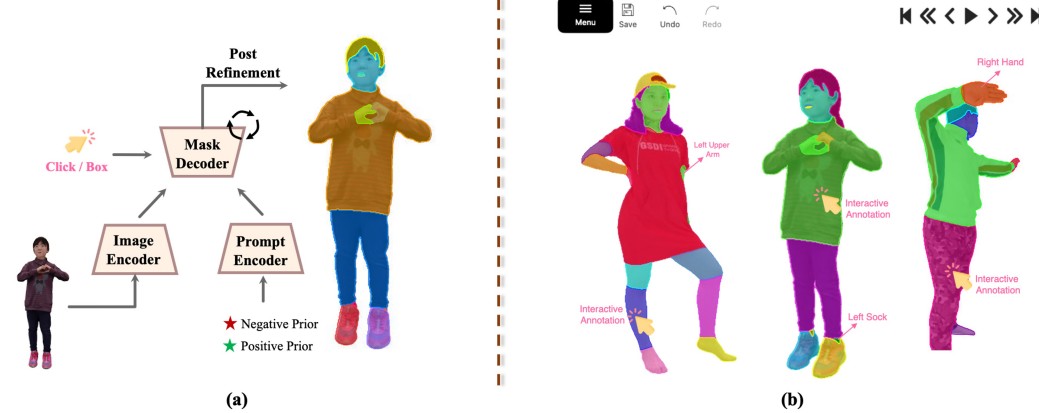

Figure 8: The dataset construction pipeline. (a) The instance and semantic segmentation annotation pipeline allows us to repeatedly reuse the features of input images and the prior negative and positive coordinates. (b) Accelerating annotation procedure by leveraging the Segment Anything Model  [95].

## A.2    More Training Details

To accelerate the training process, we employ Flash-Attention-v2 [96] from the `xFormers` library [97], gradient checkpointing [98], and BFloat16 mixed-precision arithmetic [99]. Leveraging a pre-trained model and human geometric priors, our method takes 7 days of training on 8 NVIDIA A800 GPUs.

**Differentiable 3DGS Rasterization.**    A modified 3DGS rasterization implementation[1] [100] supports depth, alpha, and **normal** rendering. Additionally, we extend 3DGS rasterization pipeline to incorporate feature attributes, enabling **feature** rendering from new perspectives, where all operations are differentiable. Due to limitations in GPU memory, we render 3DGS features $\mathbf{f}$ with a dimensionality of up to 1024 at most. Initially, we filter out low-opacity 3D Gaussian splats ($\sigma_p < 0.005$) to enhance rendering speed without compromising quality.

---

[1] https://github.com/BaowenZ/RaDe-GS.git

# B More Details of HUMANCRAFTER

**Dataset Normalization.** To better learn the 3DGS attributes, we place the human scans at the origin of the coordinate system and normalize them to a unit cube, so that they are located within the bounding box ($[-1, 1]^3$) in the world space. The camera poses of the rendered views are normalized with a global scale of 1.4.

**3D Gaussian Primitives Normalization.** As 3D Gaussians are unstructured explicit representation , the parameterization of the output parameters can affect the model's convergence. For numerical values of Gaussian splat properties, we set to Spherical Harmonics to 3, and all attributes confined within $[0, 1]$ in preparation of diffusion-based generation, outputs of 3DGS are all activated by the `sigmoid` function , except for $\mathbf{r}$, which is $L_2$-normalized to yield unit quaternions. RGB color $c$ and opacity $o$ are already supposed to be in $[0, 1]$. Raw scale $\hat{\mathbf{s}}$ is linearly interpolated with predefined values $s_{\min}$ and $s_{\max}$ [65]. $\mathbf{s} := s_{\min} \cdot \texttt{sigmoid}(\hat{\mathbf{s}}) + s_{\max} \cdot (1 - \texttt{sigmoid}(\hat{\mathbf{s}}))$. Here, $s_{\min}$ and $s_{\max}$ are set to `5e-4` and `2e-2` respectively to represent fine-grain details.

**Ablation on Image Encoder [20].** We freeze the image encoders based on DINOv2's best practices to leverage its pre-trained features and to maintain training efficiency by only training a lightweight decoder. We validated this design choice with an ablation study on the 2K2K dataset, as shown in Table 5. The results indicate that fine-tuning the image encoder provides only marginal gains (+0.032 PSNR).

Table 5: Ablation study on the effect of fine-tuning the DINOv2 image encoder. The experiment is conducted on the 2K2K dataset.

| Method | PSNR ↑ | SSIM ↑ | LPIPS ↓ |
|---|---|---|---|
| Fine-tuned DINOv2 | **23.521** | **0.918** | 0.046 |
| Frozen DINOv2 | 23.489 | 0.916 | **0.045** |

**Ablation on Dual-Transformer.** The first Transformer (Feature Aggregation) focuses on the general task of aggregating multi-view geometric and appearance features. The second Transformer (Pixel-align Aggregation) is specialized, using an attention mechanism to translate these fused features into the structured parameters of our semantic 3D Gaussians. To validate this, we trained a baseline with a single, monolithic Transformer, and the results, presented in Table 6, confirm our design is superior.

Table 6: Ablation study of our two-stage Transformer architecture. The baseline "w/o Pixel-align Aggregation" uses a single, monolithic Transformer.

| Architecture | PSNR ↑ | SSIM ↑ | LPIPS ↓ |
|---|---|---|---|
| w/o Pixel-align Aggregation | 21.183 | 0.891 | 0.067 |
| Full Model (Ours) | **23.489** | **0.916** | **0.045** |

