# OpenReview forum: "HumanCrafter: Synergizing Generalizable Human Reconstruction and Semantic 3D Segmentation"
_NeurIPS.cc/2025/Conference — NeurIPS 2025 poster_

### Official Review · Reviewer_GaxS · 2025-06-27

**Clarity:** 3
**Significance:** 2
**Originality:** 2
**Rating:** 5
**Confidence:** 2

**Summary:**

This paper introduces Human3R, a method that performs feed-forward human reconstruction and human-part semantic segmentation from a single image. The main components of the approach are: (1) Given a single image, a diffusion model performs novel view synthesis; (2) Using an estimated 3D human mesh as guidance and camera Plucker embeddings, pose-conditioned images are created; (3) A Transformer can then aggregate those multi-view pose-conditioned images and unproject them to the 3D Gaussians; (4) From the input image, some features are extracted using DINOv2. Those features are linearly combined by leveraging the attention weights from the previous network. (5) Finally, the rendering can be done for colors and semantics using a rendering function with shared weights.

Experiments show that this approach obtains SOTA results in both 3D segmentation and reconstruction.

**Questions:**

I would like to hear from the authors on the following points:
- **Novelty:** What are the main contributions and insights of this paper? From my understanding, it looks like the proposed approach is mainly an aggregation of techniques proposed in prior works.
- **Technical detail** How are the depth maps and offsets predicted in the feature aggregation component?
- **Evaluation:** How can we perform 3D segmentation with Sapiens (2 input images)? How can Human3R handle more than one image?
- **Ablations:** Could models in the spirit of DUSt3R provide a better initialization?

**Ethical Concerns:**

["NO or VERY MINOR ethics concerns only"]

**Final Justification:**

After reviewing the author's rebuttal, I no longer have concerns related to a lack of novelty or insights. Additionally, answers to my technical questions allowed me to understand the method and baselines better. Weaknesses highlighted by other reviewers were also addressed during the rebuttal.

I think this paper presents a nice research idea (using the synergy between 3D reconstruction and segmentation); the method relies on the latest advances in the field of 3DGS features fields and human reconstruction, and is clearly explained. Comparisons to the SOTA approaches and ablation studies validate the approach on multiple widely used datasets. I increase my final rating to "Accept".

**Limitations:**

Yes.

**Paper Formatting Concerns:**

The Appendices should be given as supplementary materials and not be included in the main paper.

**Quality:**

3

**Strengths And Weaknesses:**

# Strengths

The proposed approach successfully extends 3DGS Feature Field to humans. Even if the idea of adding feature prediction is not new, applying such a method to humans is not straightforward.

Human3R obtains SOTA performance in both 3D human segmentation and reconstruction.

The authors will release the code and the introduced dataset.


# Weaknesses

The overall writing of the paper could be polished (unclosed parenthesis L68, "Our approach is builds upon", missing capital letter L251, inconsistent sentences such as L243, ...).

I feel like the method is an aggregation of many "recipes" that worked in other works. Specifically, multi-view images are generated using SV3D [65], and the SMPL guidance with side-view normal image follows works like Sifu [69]. The cross-view attention module does not seem different from prior large reconstruction models. The use of DINOv2 features and their aggregation also seems very close to LUDVIG [26]. The training loss is in the same spirit as prior works such as LSM [27]. I do not argue that the paper is not novel at all as elements come from many different papers, and the application is new, but I wonder what insights this paper gives, and how the community can build on top of it.

Additionally, I have the following doubts (these points could probably be clarified):
- L143: How are the depth maps and offsets predicted?
- Table 1: How is the Sapiens [15] baseline computed? I understand how it can work when combined with Human3Diffusion [47], but I struggle to understand how it could alone predict the 3D segmentation given 2 images.
- Table 1: How can Human3R support 2 input views?
- L265: If the ability to establish correspondences, why not use recent models like DUSt3R [a], VGGT [b], or even DUNE [c] (that is also capable of predicting 3D human meshes)?



[a] Wang, Shuzhe, et al. "Dust3r: Geometric 3d vision made easy." Proceedings of the IEEE/CVF Conference on Computer Vision and Pattern Recognition. 2024.\
[b] Wang, Jianyuan, et al. "Vggt: Visual geometry grounded transformer." Proceedings of the Computer Vision and Pattern Recognition Conference. 2025.\
[c] Sarıyıldız, Mert Bülent, et al. "DUNE: Distilling a Universal Encoder from Heterogeneous 2D and 3D Teachers." Proceedings of the Computer Vision and Pattern Recognition Conference. 2025.

# Overall recommendation

This paper presents an approach that performs human reconstruction and 3D segmentation. The authors leverage the latest advances in 3DGS that add a feature field and leverage a 2D foundation model, which benefits both tasks. Despite those strengths, I feel like the writing could be polished, and I wonder if the method is technically novel enough for NeurIPS. Still, I rate this paper as "Borderline Accept" as it is mostly clear, the application of 3DGS Feature Field is novel in the human field, and the proposed approach obtains good results.

---

> ### Author Rebuttal · Authors · 2025-07-31
>
> We sincerely thank the reviewer for the positive feedback and insightful summary. We appreciate the recognition of our work's key strengths: the non-trivial extension of 3DGS Feature Fields to humans, the state-of-the-art performance achieved, and our commitment to releasing the code and dataset to foster future research in the community. We sincerely hope our responses below adequately address your concerns.
>
> **Q1: Polish Writing.**
>
> Thank you for your careful and detailed review. We will correct all typos and inconsistencies in the revised manuscript, including the specific points you raised:
> - L68: The missing parenthesis ")" will be corrected.
> - L252: The capitalization of "We" will be fixed.
> - L243: The inconsistent sentence will be rewritten for clarity.
> Furthermore, we will conduct a thorough proofread of the entire paper to improve its overall polish.
>
> **Q2: Clarification of the Novelty**
>
> We appreciate the opportunity to clarify our motivation and contributions.
> Our primary insight and contribution is beyond it, as an unified pipeline `synergizing 3D reconstruction and semantic segmentation`. This proposed philosophy can unlock many novel and practical downstream applications, such as semantic-guided 3D reasoning, context-aware human behavior analysis, and interactive semantic editing in immersive AR/VR environments, `which have not yet been thoroughly explored`. To achieve this goal, we first devote substantial effort to `curating data-label pairs`, recognizing the extreme scarcity of labeled 3D datasets. Furthermore, we are pleased to observe the following advantageous properties emerging from our unique design:
>
> 1.  **$\color{#8B2E22}{\textbf{Segmentation can benefit reconstruction}}$:** We leverage `Pixel-Align Aggregation` to enhance 3D reconstruction by incorporating semantic information on the 2K2K Dataset. Specifically, we utilize two types of semantic information: (1) pre-trained DINOv2 features and (2) manually annotated semantic maps. Both have been shown to improve reconstruction quality, as semantic segmentation helps the model prioritize specific parts and clothing regions.
>
> | Method | DINOv2  | Semantic Guidance | PSNR ↑ | SSIM ↑ | LPIPS ↓ |
> | :--- | :---: | :---: | :---: | :---: | :---: |
> | Base Model (`w/o Pixel-Align Aggregation`)| ❌ | ❌ | 21.183 | 0.891 | 0.067 |
> | Feature Distillation | ✅ |  ❌ | 22.464 | 0.896 |  0.055 |
> | Ours (Full Model) | ✅ | ✅ | 23.489 | 0.916 | 0.045 |
>
>
> 2.  **$\color{#8B2E22}{\textbf{Reconstruction  can benefit segmentation}}$:** Limited segmentation data often struggles to achieve convergence for 3D consistency segmentation maps. To address this, we naturally incorporate reconstruction priors, which significantly enhance 3D consistency and improve mIOU from `0.724` to `0.840` **(+16%)**. Notably, on human mesh datasets, our approach achieves better `3D consistency and coherence` compared to 2D foundation models (e.g., Sapiens).
>
> Additionally, our ablation studies validate the impact of key design components—such as loss functions and geometry guidance. We will release the entire pipeline (code and constructed dataset) to support community adoption.
>
> **Q3: Technical Details (Depth/Offset Prediction, Sapiens/Human3R in Table 1).**
>
> - Depth/Offset Prediction: Inspired by Splatter Image [1], the depth maps and 2D offsets are predicted by a small, two-branch MLP head attached to the output of our first (feature aggregation) Transformer. This head regresses per-pixel depth and offset values, which are then used to unproject pixels into an initial 3D XYZ.
> - Sapiens + Human3Diffusion (2-view baseline): To create a strong baseline for two-view 3D segmentation, we adapted the official open-source implementation of Human3Diffusion. Since their framework already uses a multi-view diffusion model and a 3D Gaussian Splatting representation similar to ours, we "hacked" their 3DGS renderer to support feature rendering. This allowed us to establish a baseline with a comparable architecture. As shown in Table 1, Human3R achieves **higher reconstruction fidelity with a lower inference time**, demonstrating the effectiveness of our integrated approach.
> - Human3R (2-view support): We simply bypass the diffusion-based view generation and feed the second ground-truth view directly into our feature aggregation module. This highlights our framework's flexibility. We will add these implementation details to the experimental setup section in our revision.
>
> **Q4: Using newer correspondence models (e.g., DUSt3R).**
>
> This is an excellent and insightful suggestion. We would like to clarify that our primary baseline, LSM, is already built upon DUSt3R. As mentioned in lines 226-227 of our paper, we fine-tuned it on our human dataset to ensure a fair and direct comparison.
>
> Regarding the newer models you mentioned (VGGT [2], DUNE [3]), they were not yet established or widely adopted at the time we were conducting our primary experiments, and integrating them would require substantial effort to adapt and fine-tune for our specific downstream task. We thank the reviewer for this idea and will add it to our future work/discussion section.
>
> **Q5: Appendices in supplementary material.**
>
> Thank you for careful review. We will move all appendices to the supplementary material for the camera-ready version, as per NeurIPS formatting guidelines.
>
>
> ---
> **Reference:**
>
> [1] Splatter Image: Ultra-Fast Single-View 3D Reconstruction
>
> [2] VGGT: Visual Geometry Grounded Transformer, CVPR 2025
>
> [3] DUNE: Distilling a Universal Encoder from Heterogeneous 2D and 3D Teachers, CVPR 2025

---

> > ### Comment · Reviewer_GaxS · 2025-08-04
> >
> > Thanks to the authors for the rebuttal, which addresses most of my concerns, especially related to novelty/insights and technical details. I do not have further questions following the rebuttal.

---

> > > ### Author Response · Authors · 2025-08-04
> > >
> > > Dear Reviewer **GaxS**:
> > >
> > > We are pleased to hear that our responses have addressed your concerns. We truly appreciate your constructive comments throughout the review process, which have greatly helped in improving our work!
> > >
> > > **Sincerely**,
> > >
> > > The Authors

---

### Official Review · Reviewer_wsxq · 2025-06-29

**Clarity:** 2
**Significance:** 3
**Originality:** 3
**Rating:** 4
**Confidence:** 3

**Summary:**

This paper proposes a novel feed-forward end-to-end pipeline to reconstruct human geometry, appearance, and semantic features from a single image, using 3DGS as the representation, allowing efficient reconstruction and inference and many downstream applications like AR/VR.

**Questions:**

1. I don't understand why the authors introduce two transformer modules to predict the 3DGS from multi-view images (if I understand correctly, one for feature aggregation and another for predicting the Gaussians) instead of directly using a single module/transformer. Could the authors elaborate more on the benefits of such a design?
2. While training an end-to-end reconstruction model is efficient, such a data-driven approach may lack generalizability and result in poorer reconstruction quality. I wonder if the authors could elaborate on why they believe an end-to-end model is preferable to optimization-based methods. For example, they could provide comparison results with those obtained by directly optimizing a 3DGS from multi-view images generated by the diffusion model and the per-image semantic feature fields.
3. How is the SMPL model used in the proposed method? Is it provided as input or estimated from some human mesh recovery model?

**Ethical Concerns:**

["NO or VERY MINOR ethics concerns only"]

**Final Justification:**

After the rebuttal, the authors addressed most of my concerns, especially the motivation and the significance of the proposed method. Although the writing of the current version of the paper still lacks clarity, I appreciate the value of jointly learning reconstruction and semantics claimed in this paper and incline towards acceptance.  I suggest the authors updating the introduction to clarify the motivation of this work and the method section to make the key technical contribution (i.e the feature aggregation transformer and the pixel-align aggregation transformer) clearer.

**Limitations:**

The authors mention using web-scale human datasets to train the model in their future work, which may raise potential ethical concerns.

**Quality:**

2

**Strengths And Weaknesses:**

Strengths:
1. The idea of end-to-end reconstruction is promising, with good performance and qualitative results.
2. Monocular reconstruction from a single image is still an open-ended and highly challenging task, especially for humans. It is beneficial to the community to push this field forward with new methods.

Weaknesses:
1. Unclear Motivation: In the introduction, the authors primarily discuss the limitations of existing approaches that utilize monocular videos or sparse multi-view inputs. However, the paper fails to provide a convincing motivation for using monocular images as inputs instead of addressing these limitations. I believe monocular reconstruction from a single image is a highly ill-posed problem and could be even harder to solve.
2. Rely on a 2D diffusion model. I believe the most crucial component of the proposed method is the pre-trained diffusion model, which can generate 3D consistent multi-view images from a monocular input and addresses the ill-posed 2D-to-3D problem. Therefore, the performance of the method would largely depend on the quality of the diffusion model, which is not this paper's contribution.
3. Lack of novelty: The paper proposes a pipeline to first use 3D priors to lift the input 2D image to 3D, but each module in the proposed pipeline either directly uses a pretrained model (e.g, the 2D diffusion model) or follows existing works (e.g., feed forward pixel-aligned 3DGS) without too much specific model design or motivation on why such design of combing existing techniques would be helpful to the final result. This makes the technical contribution weak.

---

> ### Author Rebuttal · Authors · 2025-07-31
>
> We thank Reviewer wsxq for the critical and detailed feedback, which has prompted us to conduct several new experiments to validate our core claims.
>
> **Q1: Why task monocular images as inputs ?**
>
> Thank you for this important question! We agree that monocular 3D reconstruction is a highly ill-posed and challenging problem. However, we believe its difficulty is precisely what makes it a fundamental and valuable area of research, for two main reasons:
>
> 1. Greater Flexibility and Efficiency: Inspired by HumanSplat and PShuman [1,2], a single-image to 3D pipeline offers significant flexibility. It allows users to reconstruct a full 3D representation from just one picture, which is essential for many downstream applications in fields like gaming, movies, fashion, and AR/VR, where multi-view data is often unavailable.
>
> 2. A Unified and Extensible Framework: The architecture is modular, allowing us to adapt it for sparse or multi-view inputs by replacing specific components (e.g., using a multi-view spml estimation model). In fact, our model can be extended to handle two input views by simply replacing the generated latent  with those from the GT input latents (encode input images via a VAE model), as shown in our experiments (Tables 1 and 3). This shows that our single-view approach provides a adaptable foundation for sparse input scenarios.
>
>
> **Q2:  Reliance on Diffusion and Lack of Novelty.**
>
> Your insight regarding the technical contribution is enlightening. We appreciate the opportunity to clarify our motivation and contributions.
>
> We group these points as they relate to the core premise and contribution of our work. We respectfully argue that our work presents significant novelty beyond a simple aggregation of techniques. The core innovation lies in the synergistic integration and non-trivial adaptation of existing components to solve the novel and challenging task of joint, single-image human reconstruction and segmentation.
>
> Our motivation is to tackle the highly ill-posed but practical single-image problem, which is now feasible due to powerful 2D generative priors. Our contribution is not the prior itself, but the framework that harnesses and lifts these 2D features into a coherent, semantic 3D representation.
>
> We appreciate the opportunity to clarify our motivation and contributions. Our primary insight and contribution is beyond it, as an unified pipeline `synergizing 3D reconstruction and semantic segmentation`. This proposed philosophy can unlock many novel and practical downstream applications, such as semantic-guided 3D reasoning, context-aware human behavior analysis, and interactive semantic editing in immersive AR/VR environments, `which have not yet been thoroughly explored`. To achieve this goal, we first devote substantial effort to `curating data-label pairs`, recognizing the extreme scarcity of labeled 3D datasets. Furthermore, we are pleased to observe the following advantageous properties emerging from our unique design:
>
> 1.  **$\color{#8B2E22}{\textbf{Segmentation can benefit reconstruction}}$:** We leverage `Pixel-Align Aggregation` to enhance 3D reconstruction by incorporating semantic information on the 2K2K Dataset. Specifically, we utilize two types of semantic information: (1) pre-trained DINOv2 features and (2) manually annotated semantic maps. Both have been shown to improve reconstruction quality, as semantic segmentation helps the model prioritize specific parts and clothing regions.
>
> | Method | DINOv2  | Semantic Guidance | PSNR ↑ | SSIM ↑ | LPIPS ↓ |
> | :--- | :---: | :---: | :---: | :---: | :---: |
> | Base Model (`w/o Pixel-Align Aggregation`)| ❌ | ❌ | 21.183 | 0.891 | 0.067 |
> | Feature Distillation | ✅ |  ❌ | 22.464 | 0.896 |  0.055 |
> | Ours (Full Model) | ✅ | ✅ | 23.489 | 0.916 | 0.045 |
>
>
> 2.  **$\color{#8B2E22}{\textbf{Reconstruction  can benefit segmentation}}$:** Limited segmentation data often struggles to achieve convergence for 3D consistency segmentation maps. To address this, we naturally incorporate reconstruction priors, which significantly enhance 3D consistency and improve mIOU from `0.724` to `0.840` **(+16%)**. Notably, on human mesh datasets, our approach achieves better `3D consistency and coherence` compared to 2D foundation models (e.g., Sapiens).
>
> Additionally, our ablation studies validate the impact of key design components—such as loss functions and geometry guidance. We will release the entire pipeline (code and constructed dataset) to support community adoption.
>
>
> **Q3:  Why two Transformers instead of a single transformer?**
>
> This is an excellent question. The first Transformer (Feature Aggregation)  focuses on the general task of aggregating multi-view geometric and appearance features. The second Transformer (Pixel-align Aggregation) is specialized, using an attention mechanism to translate these fused features into the structured parameters of our semantic 3D Gaussians. To validate this, we have conducted ablation studies with a single, monolithic Transformer, and the results confirm our design is superior:
>
> | Architecture | PSNR ↑ | SSIM ↑ | LPIPS ↓ |
> | :--- | :---: | :---: |:---: |
> | w/o Feature Aggregation Transformer | 22.203 | 0.890 | 0.064|
> | w/o Pixel-align Aggregation Transformer | 21.183 | 0.891 | 0.067|
> | Full Model (Ours) | 23.489 | 0.916 | 0.045 |
>
>
> **Q4: Why an end-to-end model instead of per-subject optimization?**
>
> Thank you for your constructive feedback. We agree that optimization-based methods can achieve more high-quality results. However, we respectfully offer a different perspective on the primary goal of our work.
>
> 1. Unlike optimization-based methods that often require minutes to hours per subject, our model is designed for efficiency and scalability , enabling 3D reconstruction in a matter of seconds. We believe this is a critical feature for many real-world applications where speed is essential.
>
> 2. Furthermore, the two approaches are not mutually exclusive. As explored in recent work like InstantSplat [3], efficient, end-to-end models can serve as an excellent initialization for optimization-based methods, significantly accelerating their convergence. This potential synergy further highlights that developing fast, feed-forward models is a valuable research direction.
>
>
> **Q5: How is the SMPL model used?**
>
> Thank you for this important question. For an input image, we first estimate SMPL(-X) parameters with PIXIE [4]. The resulting mesh provides a structural guidance for the human's pose and shape.
>
> **Q6: How is the SMPL model used?**
> We appreciate your important point.  Our dataset will be built exclusively from permissively-licensed  images (e.g., HGS-1M [5]). For any future work, we will adhere to strict ethical guidelines, including data source transparency and privacy filtering.
>
> ---
> **Reference:**
>
> [1] HumanSplat: Generalizable Single-Image Human Gaussian Splatting with Structure Priors, NeurIPS 2024
>
> [2] PSHuman: Photorealistic Single-image 3D Human Reconstruction using Cross-Scale Multiview Diffusion and Explicit Remeshing, CVPR 2025
>
> [3] InstantSplat: Sparse-view Gaussian Splatting in Seconds, Arxiv 2024
>
> [4] Feng, Yao, et al. "Collaborative regression of expressive bodies using moderation.", 3DV 2021
>
> [5] SIGMAN: Scaling 3D Human Gaussian Generation with Millions of Assets, ICCV 2025

---

> ### Comment · Reviewer_wsxq · 2025-08-01
>
> Thank the authors for their detailed rebuttal, which addressed most of my concerns. I understand the key contribution lies in synergizing 3D reconstruction and semantic segmentation, and the additional results can support this claim. I suggest updating the introduction to clarify the motivation of this work, specifically why using monocular input, why using an end-to-end model, and why jointly learning reconstruction and semantics are important, as well as the method section to make the key technical contribution clearer.
>
> In addition, besides the good performance and the detailed ablation, it's still not clear how the performance of this work would be dependent on the image-to-3D diffusion priors in the first stage. Could the authors provide an additional ablation experiment on how different priors will affect the final result? For example, using Zero-1-to-3 [1] instead of SV3D. Is the performance independent of the prior diffusion model, or will it benefit from a stronger prior?
>
> [1] Liu et al. Zero-1-to-3: Zero-shot one image to 3d object. ICCV 2023.

---

> ### Author Response · Authors · 2025-08-04
> **Different generative priors will affect the final result?**
>
> Dear **Reviewer wsxq**,
>
> We sincerely thank you for your positive and constructive feedback. It is inspiring to hear that our previous rebuttal addressed most of your concerns. As you suggested, we will update the **Introduction section** to make our motivation and contributions clearer. Regarding your concern about "how model performance depends on image-to-3D diffusion priors":
>
> This is a very insightful question. To address it, we have made a concerted effort to conduct an **ablation study** analyzing the impact of the 2D diffusion prior on our final results.
>
> **Clarification:** Due to Zero-1-to-3's [1] single-view limitation, we instead compare SV3D against MVDream [2], both multi-view diffusion models capable of generating consistent 3D content from single images. Our findings indicate that MVDream is prone to inconsistencies in its generated multi-view outputs, whereas SV3D demonstrates superior coherence. This technical alignment ensures a meaningful comparison of `how different priors affect the final result`.
>
> We replaced SV3D with the MVdream model and evaluated the performance on the 2K2K dataset (with the same training datasets and settings). The results are presented in the table below:
>
> | 2D Diffusion Prior | PSNR ↑ | SSIM ↑ | LPIPS ↓ |#Points ↑|
> |--------------------|--------|--------|---------|--------|
> | Ours + MVdream [2] | 21.685 | 0.850  | 0.166   | `694.84` |
> |--------------------|--------|--------|---------|--------|
> | Ours + MVdream [2] | 22.531 | 0.895  | 0.049   | `5636.17` |
> | Ours + SV3D        | **23.489** | **0.916**  | **0.045**  | **`6453.23`** |
>
> Even with suboptimal priors (e.g., MVDream), our method consistently produces high-quality, robust results while reducing multi-view inconsistencies, as evidenced by the increased **number of matching points (#Points ↑)**. It indicates that the core components of our architecture extract and utilize geometric and semantic information without solely depending on a single, specific generative prior.  We will incorporate this ablation study and its analysis into revision.
>
> Once again, we thank you for your time and for providing this valuable feedback. It has helped us significantly improve the clarity and completeness of our work.
>
> Sincerely,
> The Authors
>
> [1] Liu et al. Zero-1-to-3: Zero-shot one image to 3d object. ICCV 2023.
>
> [2] Shi et al. MVDream: Multi-view Diffusion for 3D Generation. ICLR 2024

---

> > ### Comment · Reviewer_wsxq · 2025-08-04
> >
> > Thank you for the response and providing the ablation results on different diffusion priors. I have no further questions and will raise my rating towards accept.

---

### Official Review · Reviewer_dwcT · 2025-07-01

**Clarity:** 3
**Significance:** 3
**Originality:** 3
**Rating:** 5
**Confidence:** 5

**Summary:**

This paper proposes Human3R, a unified network to jointly reconstruct 3D human and the corresponding semantics from a single-view image input. The key idea is to use the SV3D framework to generate multiview images with a number of priors such as SMPL, side normals, etc. and lift them into 3D Gaussians. At the same time, it distills DINOv2 feature to the 3DGS, which contributes to the following segmentation. It was trained on THuman2.1, 2K2K, and Human MVImageNet datasets.

**Questions:**

See weaknesses.

**Ethical Concerns:**

["NO or VERY MINOR ethics concerns only"]

**Final Justification:**

The authors addressed my questions during the rebuttal. I lean towards accepting this submission. The idea of jointly conducting reconstruction and segmentation is valuable. I recommend revising the segmentation evaluation to make the conditions, pros, and cons of each method clearer, and incorporating the rebuttal discussion into the final version.

**Limitations:**

Yes.

**Quality:**

3

**Strengths And Weaknesses:**

Strengths
* The visualization looks promising, both reconstruction and segmentation
* Open-sourcing the trained model would be beneficial to the community
* From Tab.4, the proposed Pixel-align Aggregation contributes a lot to the performance. The approach could be helpful in many related 3D tasks.

Weaknesses:
* The results lack failure cases for in-the-wild settings. I'm curious when and how it fails for in-the-wild images.
* Lack of comparison with general 3D generation, which shows strong ability in humans as well, e.g. Hunyuan3D, SPAR3D

Minor:
* Less clear presentation of the contribution. Since each step is not new to the field, what makes your method different the priors are not very well-highlighted in the writing. I'm sure there's a lot of original work based on the results provided, but please state them clearly.
* Segmentation should be evaluated in third-party datasets, as the methods compared with are general.

---

> ### Author Rebuttal · Authors · 2025-07-31
>
> We thank the reviewer for their positive feedback and for recognizing the qualitative results and the effectiveness of our `Pixel-align Aggregation module`. We appreciate the opportunity to contribute to the community and we have provided detailed clarifications and responses below to address your concerns.
>
> **W1: Lack of discussion on failure cases.**
>
> We thank the reviewer for this suggestion! While HUMAN3R demonstrates robust generalization across various challenging scenarios, the model's performance may degrade under two primary conditions:
>
> - Extreme Poses: For extreme poses that are challenging for pose estimation models [1], our model may generate minor misalignments in fine details (e.g., in the arms). However, the overall limb structure remains intact, a robustness afforded by the geometric prior.
>
> - Low-Quality Inputs: When processing low-resolution or heavily compressed images, the input may lack the necessary high-frequency information for high-fidelity reconstruction. It leads to a degradation of fine-grained details, such as facial features or clothing textures in the outputs.
>
> We will include this in the "Limitations and Failure Cases" section. Thank you for your valuable feedback!
>
>
> **W2: Comparison with general 3D generation models.**
>
> We thank the reviewer for their constructive feedback!
>
> First, we would like to clarify that our method was compared against recent 3D generation models, including LGM, GRM, and InstantMesh (which is the base model for Hunyuan3D 1.0). To ensure a fair comparison, we fine-tuned both the LGM and GRM models on our dataset.
>
> Furthermore, to fully address your concern, we have benchmarked our method against Stability AI's recent model, SPAR3D, on the test set of the 2K2K dataset. The results below further demonstrate the effectiveness of our approach.
>
> | Method | PSNR ↑ | SSIM ↑ | LPIPS ↓ |
> | :--- | :---: | :---: | :---: |
> | SPAR3D | 20.295 | 0.854 | 0.063 |
> | Ours | 23.489 | 0.916 | 0.045 |
>
> We will incorporate these new results into the revised version of the paper. Thank you for your valuable suggestion.
>
> **W3: Clarify the contribution.**
>
> We appreciate the opportunity to clarify our motivation and contributions.
>
> Our primary insight and contribution is beyond it, as an unified pipeline `synergizing 3D reconstruction and semantic segmentation`. This proposed philosophy can unlock many novel and practical downstream applications, such as semantic-guided 3D reasoning, context-aware human behavior analysis, and interactive semantic editing in immersive AR/VR environments, `which have not yet been thoroughly explored`. To achieve this goal, we first devote substantial effort to `curating data-label pairs`, recognizing the extreme scarcity of labeled 3D datasets. Furthermore, we are pleased to observe the following advantageous properties emerging from our unique design:
>
> 1.  **$\color{#8B2E22}{\textbf{Segmentation can benefit reconstruction}}$:** We leverage `Pixel-Align Aggregation` to enhance 3D reconstruction by incorporating semantic information on the 2K2K Dataset. Specifically, we utilize two types of semantic information: (1) pre-trained DINOv2 features and (2) manually annotated semantic maps. Both have been shown to improve reconstruction quality, as semantic segmentation helps the model prioritize specific parts and clothing regions.
>
> | Method | DINOv2  | Semantic Guidance | PSNR ↑ | SSIM ↑ | LPIPS ↓ |
> | :--- | :---: | :---: | :---: | :---: | :---: |
> | Base Model (`w/o Pixel-Align Aggregation`)| ❌ | ❌ | 21.183 | 0.891 | 0.067 |
> | Feature Distillation | ✅ |  ❌ | 22.464 | 0.896 |  0.055 |
> | Ours (Full Model) | ✅ | ✅ | 23.489 | 0.916 | 0.045 |
>
>
> 2.  **$\color{#8B2E22}{\textbf{Reconstruction  can benefit segmentation}}$:** Limited segmentation data often struggles to achieve convergence for 3D consistency segmentation maps. To address this, we naturally incorporate reconstruction priors, which significantly enhance 3D consistency and improve mIOU from `0.724` to `0.840` **(+16%)**. Notably, on human mesh datasets, our approach achieves better `3D consistency and coherence` compared to 2D foundation models (e.g., Sapiens).
>
> Additionally, our ablation studies validate the impact of key design components—such as loss functions and geometry guidance. We will release the entire pipeline (code and constructed dataset) to support community adoption.
>
> **W4: Why not evaluate the 3D segmentation tasks on third-party datasets?**
>
> We agree that evaluation of diverse datasets is important. However, to the best of our knowledge, our work was the first work to explore a large-scale,  `3D human dataset with detailed, multi-class part-segmentation labels (28 classes)`. This is precisely why we invested significant effort in curating our dataset.
> We believe our released dataset will serve as a valuable new benchmark for the community and help drive future research in this domain.
>
> ---
> **Reference:**
>
> [1] Feng, Yao, et al. "Collaborative regression of expressive bodies using moderation.", 3DV 2021
>
> [2] SPAR3D: Stable Point-Aware Reconstruction of 3D Objects from Single Images, arXiv 2025
>
> [3] SiFU:Side-view Conditioned Implicit Function for Real-world Usable Clothed Human Reconstruction, CVPR 2024
>
> [4] HumanSplat: Generalizable Single-Image Human Gaussian Splatting with Structure Priors, NeurIPS 2024
>
> [5] PSHuman: Photorealistic Single-image 3D Human Reconstruction using Cross-Scale Multiview Diffusion and Explicit Remeshing, CVPR 2025

---

> > ### Comment · Reviewer_dwcT · 2025-08-03
> >
> > Thanks for the response. Regarding the segmentation comparison in Figure 3, it's quite abnormal to see a large black region for Sapiens. LSM has similar artifacts as well. Since Sapiens is a 2D model, my understanding is that the target GT views are fed to Sapiens directly to obtain the segmentation. Could you elaborate on the reason?

---

> ### Author Response · Authors · 2025-08-04
> **Response to Segmentation Artifact Inquiry**
>
> Thank you for your detailed observation regarding the segmentation artifacts in Figure 3. This is a good question. Thank you for bringing it up for discussion. We clarify these phenomena as follows:
>
> - **Sapiens' Black Regions:** As a purely 2D image-based model pre-trained on natural images, Sapiens exhibits a `domain gap` when applied to `rendered human mesh datasets`. For instance, it misclassifies yellow/white clothing regions as "background" or "other categories", or shows degraded segmentation performance for back views due to the long-tail data distribution.
>
> - **LSM's Black Regions:** These occur primarily in (1) `textureless regions` (e.g., solid-color clothing) where feature matching fails and (2) input views with large camera distances that cause failure of LSM's matching mechanism and subsequent performance degradation.
>
> However, this can be compensated by `our human body prior`, resulting in better performance through SMPL(-X) parameters. For instance, in Fig 3 row 3, our method can still recover **reasonable segmentation** outcomes by leveraging the human body prior.
>
>
> We will include additional segmentation results ( 360° camera trajectory) of three methods in the revised manuscript and supplementary materials to clarify the observed phenomena. Thank you again for your time and valuable input!
>
> Sincerely,
> The Authors

---

### Official Review · Reviewer_as2P · 2025-07-04

**Clarity:** 3
**Significance:** 3
**Originality:** 3
**Rating:** 5
**Confidence:** 4

**Summary:**

This paper presents a unified framework for 3D human reconstruction, novel view synthesis, and semantic body-part segmentation, from a single image. Built upon 3DGS, the proposed method integrates human appearance prior from SV3D and 2D image features (CLIP image encoder, Dinov2) to achieve semantically consistent 3D representations.
The network consists of two core Transformer modules. The first Tranformer aggregates 3D features and human geometry across source views into pixel-aligned 3D point maps,. The second one applies an attention-based mechanism to project these features into a set of semantic 3D Gaussians, each encapsulating geometry, appearance, and semantic cues. By representing outputs as Gaussians and defining a differentiable rasterizer, the method allows direct rendering of feature maps (including RGB, depth, or segmentation maps) from novel views, supporting a multi-task supervision strategy without the need for per-subject optimization or fine-tuning.
Experiments are conducted on 2K2K & THuman2.1 to compare human 3D segmentation & 3D reconstruction. The proposed method achieves better numeric performance than previous methods, e.g. Human3Diffuison, Sapiens, on these benchmarks. In addition, exciting qualitative results are shared.

**Questions:**

See weakness.

**Ethical Concerns:**

["NO or VERY MINOR ethics concerns only"]

**Final Justification:**

The results look promising and the proposed designs have been validated in the paper and rebuttal. Issues I observed have been properly solved in rebuttal. Therefore, I raise the final score.

**Limitations:**

Yes.

**Quality:**

4

**Strengths And Weaknesses:**

Strengths.
1. A unified framework.
Unlike traditional two-stage approaches, the proposed method, HUMAN3R, offers a single forward-pass solution for 3D reconstruction, semantic segmentation, and novel-view synthesis, all within a Transformer-based pipeline.
Also, the design of two transformer blocks—one for pixel-aligned point regression, and another for semantic 3D Gaussian synthesis—is both modular and generalizable, enabling flexible integration of 2D priors and geometric reasoning.
2. Encouraging results.
Quantitative and qualitative results on benchmarks and Internet images are quite promising.

Weakness.
1. Unclear details.
In experiments section, the paper shows that they use PIXIE as the SMPL paramters estimator. While PIXIE estimates SMPL-X parameters. Different from SMPL, SMPL-X also describe the facial expresion and hand gesture. As shown in Fig. 4, the proposed method achieves better quality at the face and hand part. It would be a surprise if only body parameters are used.
Also, very limited information about the "In-the-wild Dataset" they collected for testing, e.g. image source, standard to select the image, how to semgent the human region, etc.
2. unvalided designs.
The reason of freezing the 2 image encoders has not been properly explained. So a reasonable guess would be whether train the whole model togather would achieve further improvement.
3. Missing qualitative results on evaluation benchmarks.
To further show the superior of the proposed method, qualitative results on evaluation benchmarks would be helpful to directly demonstrate the advance part of the proposed method.

---

> ### Author Rebuttal · Authors · 2025-07-31
>
> We thank the reviewer for the insightful and thorough feedback and for acknowledging our core contributions, which introduce a unified and single-pass Transformer framework for joint 3D reconstruction, segmentation, and synthesis. Below, we provide clarifications addressing your concerns.
>
> **W1: The technical details of PIXIE [1]?**
>
> Thank you for this sharp observation! The reviewer is correct that HUMAN3R utilizes PIXIE [1] to estimate parameters for the `SMPL-X model`. The high-fidelity results for the hands and face benefit from the expressive priors provided by SMPL-X. We have corrected "SMPL" to "SMPL-X" in the revision.
>
> **W2: Details of the In-the-wild dataset?**
>
> We apologize for this omission. Our in-the-wild test set contains 8 images from public CC0 sources, selected to test generalizability under challenging scenarios (e.g., diverse poses and identities). The foreground human masks are automatically generated using the `Segment Anything Model (SAM)`[2] .
> We will add a subsection to the supplementary materials with these details.
>
> **W3: Why freezing image encoders?**
>
> Thank you for this important question. We freeze the image encoders based on DINOv2 [3]'s best practices to leverage its pre-trained features and  to maintain the training efficiency by only training a lightweight decoder. We validated the design choice with an ablation study on the 2K2K dataset, which shows that fine-tuning the image encoder only provides marginal gains (+0.032 PSNR) in our setting.
>
> | Method | PSNR ↑ | SSIM ↑ | LPIPS ↓ |
> | :--- | :---| :--- | :--- |
> | Fine-tuned DINOv2 | 23.521 | 0.920 | 0.046 |
> | Frozen DINOv2  | 23.489 | 0.916 | 0.045 |
>
> We will incorporate it into the ablation study in the revision.
>
> **W4: Missing qualitative results on evaluation benchmarks.**
>
> We thank the reviewer for this constructive feedback.  However, according to `NeurIPS 2025 policy, we are prohibited from providing images, videos, or external links during the rebuttal period.`
>
> To ensure `clarity and transparency`, we summarize our qualitative results: Qualitatively, on the THuman2.1 and 2K2K benchmarks, our method excels at capturing  details (e.g, face and clothing), yielding higher-fidelity results compared to the overly smooth or blurry surfaces from baseline methods. We are committed to adding  qualitative comparisons in the revision.
>
> ---
> **Reference:**
>
> [1] Feng, Yao, et al. "Collaborative regression of expressive bodies using moderation.", 3DV 2021
>
> [2] Kirillov, Alexander, et al. "Segment anything.", ICCV 2023
>
> [3] DINOv2: Learning Robust Visual Features without Supervision, TMLR 2024

---

> > ### Comment · Reviewer_as2P · 2025-08-07
> >
> > Thanks for your feedbacks. Issues I found have been properly resolved. I have raised the final score.

---

### Decision · Program_Chairs · 2025-09-17

**Decision:**

Accept (poster)

**Comment:**

This submission receives 3 accepts and 1 borderline accept. During the rebuttal discussion period, all reviewers' concerns have been well addressed. As stated by reviewers, monocular reconstruction from a single image is still an open-ended and highly challenging task and this submission proposes a single forward-pass solution utilizing the synergy between 3D reconstruction and segmentation. Experiments have demonstrated the effectiveness of the proposed method. The recommendation is acceptance. And authors are encouraged to include the clarification and update covered in the rebuttal discussion period to the revision.